# Non-Apoptotic Programmed Cell Death as Targets for Diabetic Retinal Neurodegeneration

**DOI:** 10.3390/ph17070837

**Published:** 2024-06-26

**Authors:** Yingjia Lin, Shuping Ke, Weiqing Ye, Biyao Xie, Zijing Huang

**Affiliations:** 1Joint Shantou International Eye Center of Shantou University and The Chinese University of Hong Kong, Shantou 515041, China; linyj@jsiec.org (Y.L.); ksp@jsiec.org (S.K.); ywq@jsiec.org (W.Y.); xby@jsiec.org (B.X.); 2Fifth Clinical Institute of Shantou University Medical College, Shantou 515041, China

**Keywords:** diabetic retinal neurodegeneration, programmed cell death, necroptosis, pyroptosis, ferroptosis, PANoptosis

## Abstract

Diabetic retinopathy (DR) remains the leading cause of blindness among the global working-age population. Emerging evidence underscores the significance of diabetic retinal neurodegeneration (DRN) as a pivotal biomarker in the progression of vasculopathy. Inflammation, oxidative stress, neural cell death, and the reduction in neurotrophic factors are the key determinants in the pathophysiology of DRN. Non-apoptotic programmed cell death (PCD) plays a crucial role in regulating stress response, inflammation, and disease management. Therapeutic modalities targeting PCD have shown promising potential for mitigating DRN. In this review, we highlight recent advances in identifying the role of various PCD types in DRN, with specific emphasis on necroptosis, pyroptosis, ferroptosis, parthanatos, and the more recently characterized PANoptosis. In addition, the therapeutic agents aimed at the regulation of PCD for addressing DRN are discussed.

## 1. Introduction

With socioeconomic development, the incidence of diabetes mellitus continues to rise. By 2045, the global prevalence of diabetes is projected to reach 693 million [1], with the highest incidence in China and India [2]. Diabetes affects multiple organs throughout the body, with diabetic retinopathy (DR) being the most common microvascular complication and the leading cause of blindness in the global working-age population. Exploring the pathogenesis of DR and developing effective treatment strategies have been key concerns in the medical field [3].

DR is classified into two stages: non-proliferative diabetic retinopathy (NPDR), characterized primarily by microaneurysms and exudates, and proliferative diabetic retinopathy (PDR), characterized by retinal neovascularization. Assessment of the severity of DR is predominantly based on retinal vasculopathy. However, there are intricate and complex physical and biochemical connections among retinal vascular cells, neurons, and glial cells, which constitute the neurovascular unit [4]. In addition to vascular cells, high glucose levels affect retinal ganglion cells (RGCs), Müller cells, and microglia, leading to neurodegeneration and visual impairment in DR [4]. Retinal pigment epithelial (RPE) cells are also involved in neurodegeneration. High glucose decreases RPE cell viability, indirectly affecting the visual response properties of RGCs [5,6,7]. Diabetes triggers multiple pathophysiological mechanisms within the retina, involving alterations in genetic and epigenetic effects, elevated free radical formation, the accumulation of advanced glycation end-product, and the upregulation of vascular endothelial growth factor (VEGF) and inflammatory mediators [8]. In this regard, novel insights suggest that interventions for DR should address not only vascular dysfunction but also retinal neurodegeneration, highlighting the necessity for innovative therapeutic modalities.

The concept of diabetic retinal degeneration (DRN) has recently gained attention, which refers to the progressive degeneration of the neuroretina under diabetic conditions [9,10]. DRN is characterized by neuronal apoptosis and glial activation accompanied by molecular mediators such as inflammatory cytokines, oxidative stress, mitochondrial dysfunction, and neurotrophic factor deficiency [11]. Diabetic patients experience progressive retinal thinning and visual dysfunction, which can be assessed through retinal image analysis, visual function tests, and translational modeling [12,13]. Importantly, even in the absence of evident signs of vasculopathy, the peripapillary retinal nerve fiber layer becomes progressively thinner in diabetic patients [14]. Animal and cellular experiments also suggest that hyperglycemia may directly affect the survival of retinal neurons, indicating that DRN likely precedes microvasculopathy and may represent a pathological process independent of microvasculopathy [9]. The cellular mechanisms underlying DRN require further investigation.

Cell death is a crucial pathophysiological process during organism development and disease progression, and modulating cell death always provides a potential strategy for disease treatment. Recent studies have discovered several types of cell death occurring under certain pathological conditions, characterized by both programmed regulation and cell necrosis effects. These include pyroptosis, necroptosis, ferroptosis, lysosome-dependent cell death, autophagic cell death, and PANoptosis, among others [15,16,17]. Unlike classical apoptosis, which rarely triggers an inflammatory response, activation of these programmed cell death (PCD) pathways ultimately leads to cell lysis, the release of cellular contents, and the secretion of inflammatory mediators, thereby triggering inflammation [18]. Non-apoptotic PCD plays a critical role in maintaining tissue health and regulating disease progression [18,19] and is involved in various pathological processes such as tumors, neurodegenerative disorders, immune inflammation, and cardiovascular disease [15]. Notably, the activation of non-apoptotic PCD occurs in a programmed and modulable manner, which provides the possibility to control inflammation and intervene in diseases [20,21]. Targeting non-apoptotic PCD is expected to be a key component of future therapeutic strategies.

In recent years, a growing body of research has underscored the significance of non-apoptotic PCD in the progression of DRN. This review aims to highlight recent advancements in identifying the role of distinct non-apoptotic PCD types across diverse retinal cell populations within the diabetic retina and to discuss the exploration of therapeutic approaches based on the regulation of non-apoptotic PCD pathways as a means to treat DRN (Figure 1).

## 2. Ferroptosis

### 2.1. Overview of Ferroptosis

Ferroptosis, first defined by Scott J. Dixon in 2012, is a form of non-apoptotic PCD caused by an imbalance in intracellular iron metabolism, leading to cell death through oxidative stress and lipid peroxidation damage [22]. Several mechanisms are involved in ferroptosis, including the cystine-glutathione (GSH)–glutathione peroxidase 4 (GPX4) signaling pathway, phospholipid peroxidation, iron regulation, and cellular metabolism [23].

The GPX4 pathway is a classic signaling mechanism of ferroptosis that primarily involves regulating intracellular lipid peroxidation. GPX4 is an enzyme with antioxidant properties that catalyzes the reduction of lipid peroxides by GSH, thereby inhibiting lipid peroxidation [22,24]. Under disease conditions, the activity of GPX4 can be inhibited due to the dysfunction of its upstream regulator, system Xc-, a cystine/glutamate antiporter. This inhibition results in the accumulation of lipid peroxide, leading to irreversible membrane damage and cell death [25].

Phospholipid peroxidation plays a vital role in ferroptosis [23]. Lipid peroxidation begins with the reaction of molecular oxygen to form peroxyl radicals and then disrupts membrane integrity [26]. Notably, neuronal cell membranes are rich in polyunsaturated fatty acids, making them particularly sensitive to lipid peroxidation [26]. Two enzymes, acyl-CoA synthetase long-chain family member 4 (ACSL4) and lysophosphatidylcholine acyltransferase 3 (LPCAT3), are significant drivers of ferroptosis through phospholipid peroxidation [22].

Iron also plays an essential role in ferroptosis. Ferroptosis is regulated by the iron-dependent Fenton chain reaction, which involves the reaction between iron ions and hydrogen peroxide to produce reactive oxygen species (ROS) and initiate lipid peroxidation [23,26]. In addition, abnormal cellular metabolism contributes to the formation of phospholipid peroxides. Gao et al. found that ferroptosis can be induced by cystine starvation, requiring iron-carrier transferrin and the amino acid glutamine in serum [27]. These results suggest the importance of iron regulation and metabolism in ferroptosis.

There is increasing evidence that ferroptosis plays a significant role in various pathological processes including neurodegenerative diseases and cancer. Targeting ferroptosis holds promising application prospects in treating retinal neurodegenerative diseases.

### 2.2. Ferroptosis in Diabetic Retinal Neurodegeneration

Ferroptosis is believed to be a significant factor in retinal neurodegeneration. Acrolein-induced ferroptosis promotes defects in diabetic peripheral nerves, including DRN, which can be successfully reversed by anti-acrolein therapy or ferroptosis inhibitors [28]. In diabetic patients, the expression of ferroptosis-associated biomarkers GPX4 and GSH was significantly reduced, while lipid peroxidation and ROS were increased. Moreover, these changes were more pronounced in NPDR patients compared to PDR patients, indicating a greater activation of ferroptosis in the early stages of DR [29]. Several genes associated with ferroptosis have also been shown to be associated with DRN, including TLR4, CAV1, HMOX1, TP53, IL-1B, and ATG7 [30,31,32,33,34]. These findings suggest a genetic therapeutic strategy targeting ferroptosis-mediated DRN, although its underlying mechanisms remain unclear. Several studies have demonstrated that ferroptosis may affect various retinal cells and participate in neurodegeneration through different pathophysiological mechanisms [35,36,37,38].

#### 2.2.1. Retinal Pigment Epithelium

The RPE plays a role in antioxidant defense and barrier maintenance, and its death can impair the function of photoreceptor cells and trigger inflammation, ultimately resulting in retinal neurodegeneration. RPE has been identified as one of the main cell types undergoing ferroptosis during DRN. Increased oxidative stress-induced RPE cell dysfunction, elevated levels of ferroptosis, increased iron-mediated apoptosis, and the activation of endoplasmic reticulum stress have been observed in both in vitro and in vivo DR models [39,40]. In the early stages of diabetes, the upregulation of glial maturation factor-β, a neurodegenerative factor in the vitreous, leads to abnormal lysosomal degradation processes in RPE cells, resulting in the accumulation of ROS and ultimately inducing ferroptosis in RPE [41]. Additionally, high glucose levels induced mitochondrial dysfunction, ferritinophagy, and lysosomal instability by upregulating thioredoxin-interacting protein (TXNIP), leading to ferroptosis in RPE [42].

The dysregulation of long non-coding RNAs (lncRNAs) contributes to RPE ferroptosis in a high-glucose environment. Zhu et al. found that the downregulation of PSEN1, a circular RNA, could regulate the miR-200b-3p/cofilin-2 axis and rescue RPE ferroptosis under high-glucose stimulation [43]. Similarly, reprogramming of the miR-338-3p/SLC1A5 axis could increase the resistance of RPE to high-glucose-induced cell ferroptosis, thereby promoting RPE cell survival and function [44].

Targeting RPE ferroptosis shows promise in treating DRN. Treatment with the ferroptosis activator Erastin induced oxidative stress and cell damage in high-glucose-treated RPE cells, while the ferroptosis inhibitor Ferrostatin-1 reversed this process by activating the Xc-GPX4 axis [39]. Additionally, targeting fatty acid-binding protein 4 inhibited high-glucose-induced lipid peroxidation and ferroptosis in RPE cells through regulating the peroxisome proliferator-activated receptor-gamma (PPARγ) signaling pathway [45]. A stress-induced protein, Sestrin2, inhibited STAT3 phosphorylation and endoplasmic reticulum stress, suppressed RPE ferroptosis, and increased cell autophagy levels, thus alleviating DRN [40].

Some traditional Chinese medicines or plant extracts also showed potential in inhibiting RPE ferroptosis under diabetic conditions. Liu et al. found that the active ingredient of aromatic plant essential oils, 1,8-cineole, rescued RPE from ferroptosis by regulating the TXNIP-PPARγ signaling pathway [46]. Tang et al. discovered that Astragaloside-IV (AS-IV), a natural product extracted from Astragalus, alleviated high-glucose-induced RPE ferroptosis and oxidative stress damage by disrupting the expression of miR-138-5p/Sirt1/Nrf2 and subsequent neuronal loss [47].

#### 2.2.2. Photoreceptors

The degeneration and death of photoreceptors are significant contributors to the neurodegenerative changes in the retina. Iron overload plays a major role in photoreceptor cell death and retinal degeneration, leading to cell demise, mitochondrial dysfunction, ROS accumulation, and iron deposition [48]. Ferroptosis plays a crucial role in the pathogenesis of iron overload-induced retinal degeneration. Azuma et al. identified a mitochondrial isoform of glutathione peroxidase 4 (mGPx4), which promotes the development and survival of photoreceptors in mice, suggesting its significant role in preventing ferroptosis [49]. Some agents, such as salvianic acid A, have been shown to mitigate iron deposition, lipid peroxidation, and mitochondrial dysfunction, thereby inhibiting photoreceptor ferroptosis and retinal degeneration [48]. In addition, α-lipoic acid-L-carnitine, a bioavailable mitochondria-targeting prodrug of lipoic acid, was able to block ferroptosis and attenuate iron-induced mitochondrial dysfunction in photoreceptors, demonstrating its potential to protect against retinal degeneration and loss of photoreceptors in DR [50].

Currently, there is limited research on the role of photoreceptor ferroptosis in DRN. Gao et al. observed a significant increase in the levels of ROS, lipid peroxidation, and iron-related proteins (such as GPX4) in cultured photoreceptor cells and in the early stages of DR mice. They alleviated neurodegeneration through the ferroptosis inhibitor Ferrostatin-1, providing direct evidence for the regulation of photoreceptor ferroptosis in the treatment of DRN [51]. Further research is warranted to validate this conclusion.

#### 2.2.3. Retinal Capillary Endothelial Cells

Retinal vascular endothelial cells also undergo ferroptosis under diabetic conditions [52]. Luo et al. found that the levels of Yes-associated protein (YAP) and ROS were significantly increased in diabetic mice, while the expression of GPX4 was decreased. They demonstrated that the metabolite pipecolic acid might impede the progression of DR by inhibiting the YAP-GPX4 signaling pathway [53]. In addition, tripartite motif 46 (TRIM46), a protein involved in cellular homeostasis, could interact with GPX4 to form the TRIM46-GPX4 signaling pathway and regulate the ferroptosis of high glucose-treated human retinal capillary endothelial cells (RCECs). Inhibiting TRIM46 or maintaining GPX4 expression successfully reversed the effect of high-glucose-induced vascular hyperpermeability and inflammation [54,55].

Regulating vascular endothelial cell ferroptosis shows potential for alleviating DR. The administration of 25-hydroxyvitamin D3 significantly reduced ROS and Fe^2+^ levels while increasing levels of GSH, GPX4, and solute carrier family 7 member 11 (SLC7A11) protein in human RCECs under high glucose stimulation, thereby inhibiting ferroptosis and oxidative stress damage [56]. Inhibition of the lncRNA zinc finger antisense 1 (ZFAS1) also alleviated high glucose-induced endothelial cell ferroptosis. ZFAS1 could competitively target miR-7-5p and regulate the downstream expression of ACSL4, which is considered as a potential driver gene of ferroptosis [57]. Finally, some traditional medicine extracts, such as Amygdalin, the active ingredient of bitter almonds, also showed potential to inhibit ferroptosis, possibly by activating the Nrf2/ARE signaling pathway [58].

#### 2.2.4. Retinal Ganglion Cells

The RGC serves as a pivotal neuron in transmitting visual signals, and RGC death represents a primary hallmark of DRN. Studies have indicated a significant activation of ferroptosis in RGCs after optic nerve injury, attributed to the upregulation of 4-hydroxynonenal expression due to decreased levels of GPX4 [35]. Direct evidence for ferroptosis in RGCs during the development of DR or DRN is currently lacking. However, diabetes-related pathological mechanisms may indirectly damage RGCs and induce the occurrence of ferroptosis, as well as other forms of cell death. Ischemia-reperfusion (I/R) injury is a crucial pathological mechanism in multiple neurovascular diseases, including DRN. The inhibition of apoptosis, necroptosis, and ferroptosis all exhibited protective effects against I/R-induced RGC death, with the suppression of ferroptosis being particularly prominent [59]. Furthermore, melatonin (MT), as a promising therapeutic agent for retinal neuroprotection, inhibited RGC ferroptosis and inflammatory responses induced by retinal I/R injury, possibly by suppressing the p53 signaling pathway [60].

#### 2.2.5. Other Retinal Cells

Ryan et al. found that human induced pluripotent stem cell-derived microglia were highly susceptible to iron ions and that microglial iron overload and ferroptosis were observed in a Parkinson’s disease model. Removing microglia from the cell culture system significantly delayed iron-induced neurotoxicity. These findings highlight the role of microglial iron overload and ferroptosis in neurodegeneration [37]. In addition, the inhibition of the transforming growth factor beta (TGFβ) signaling pathway induced ferroptosis in retinal neurons and Müller cells and exacerbated retinal neurodegeneration [38]. These findings suggest that ferroptosis in both glial cells and neurons may contribute to neurodegeneration, albeit not specifically induced by diabetes.

In summary, ferroptosis, characterized by iron overload, lipid peroxidation, oxidative stress damage, and inflammatory responses, is intricately associated with retinal neurodegeneration. Various cell types have been shown to undergo ferroptosis during the pathogenesis of DRN. Targeting ferroptosis has demonstrated neuroprotective effects, offering a potential therapeutic approach for treating DRN. However, the spatiotemporal characteristics, signaling pathways, and interactions among different cells undergoing ferroptosis remain largely unclear. This poses a critical challenge that warrants further investigation before anti-ferroptosis therapy can be clinically applied for the treatment of DRN and other retinal neurodegenerative diseases.

## 3. Pyroptosis

### 3.1. Overview of Pyroptosis

Pyroptosis is a classical PCD first defined by Cookson in 2001 [61]. It serves as a crucial host defense mechanism against pathogens; however, excessive pyroptosis can lead to cytokine storms and harmful inflammation, resulting in tissue damage and organ dysfunction [62]. The NOD-like receptor protein 3 (NLRP3) is a key signaling pathway of pyroptosis, which can be activated by pattern recognition receptors (PRRs) on immune cells to scan for pathogen-associated molecular patterns (PAMPs) and damage-associated molecular patterns (DAMPs) [63,64]. NLRP3 can also be triggered by Toll-like receptors (TLRs) binding to pathogens or cytokines, leading to NF-κB activation and the release of interleukin (IL)-1β and IL-18 via caspase-1 cleavage [63]. The formation of NLRP3 inflammasome involves the recruitment of NLRP3, an adaptor protein called apoptosis-associated speck-like protein containing a CARD (ASC), caspase-1, and MAPK ERK kinase 7 (MEK7) [64,65]. Once activated, the inflammasome cleaves caspase-1, which in turn cleaves gasdermin D (GSDMD), initiating pyroptosis.

The effector proteins of pyroptosis constitute a class of pore-forming proteins known as gasdermins (GSDMs). These proteins separate the N-terminal pore-forming domain from the C-terminal inhibitory domain to form pores on the cytoplasmic membrane, leading to cell swelling and lytic cell death [66,67]. The GSDM family comprises several members, including GSDMA, GSDMB, GSDMC, GSDMD, GSDME (DFNA5), and GSDMF (PJVK/DFNB59) [68]. Pyroptosis is typically triggered by caspase-1 or caspase-11/4/5 to cleave GSDMD [66]. Under certain circumstances, caspase-3 activation can also mediate pyroptosis by cleaving GSDME [67]. It is noted that the pore formation by GSDMD is a reversible process, and pyroptosis can be halted through regulating cytoplasmic membrane repair pathways. The truly passive process of lytic cell death relies on the cytoplasmic membrane rupture mediated by a protein called Ninjurin 1 (NINJ1) [69].

Neuroinflammation is a prominent feature of neurodegeneration observed in neuro-metabolic disorders. Blocking pyroptosis by targeting the assembly of NLRP3 inflammasomes and GSDMD emerges as a potential therapeutic approach for addressing neuroinflammation and neurodegeneration-related conditions [70], including age-related macular degeneration (AMD) and Alzheimer’s disease [71]. The abnormal deposition of host proteins, such as amyloid-β, triggers neuroinflammation and neurodegeneration by activating inflammasomes as intracellular sensors of pathogens and endogenous danger signals [72]. Pyroptosis has been proposed as a potential therapeutic target for neuroinflammation and neuronal death observed in Alzheimer’s disease [73]. Meanwhile, studies have indicated that the sigma-1 receptor regulates pyroptosis and inflammation post-traumatic brain injury by regulating endoplasmic reticulum stress and calcium signaling [74]. In ophthalmology, NLRP3 inflammasome-mediated pyroptosis has been implicated in neurodegenerative disorders like glaucoma, AMD, and DR [75,76]. Certain neurosteroids, acting through sigma1 recognition sites, could influence the survival and metabolic state of neuronal and glial cells following retinal I/R injury [77]. Moreover, sigma-1 receptor ligands have demonstrated protective effects against oxidative stress-induced damage in the RPE during DRN pathology [78]. Modulating the pyroptosis signaling pathway presents a promising therapeutic strategy for retinal neurodegenerative diseases, including DRN.

### 3.2. Pyroptosis in Diabetic Retinal Neurodegeneration

Chronic inflammation plays a pivotal role in the progression of DRN. Research has demonstrated that high-glucose-induced cellular pyroptosis, particularly the release of inflammatory mediators such as IL-1β and IL-18, serves as a major source of retinal inflammation. Targeting pyroptosis and its associated inflammatory and neurodegenerative pathways provides a potential therapeutic strategy for managing DRN. In the subsequent sections, we will delineate the involvement of pyroptosis in different retinal cell types and its underlying mechanisms in the content of DRN.

#### 3.2.1. Retinal Pigment Epithelium

Several studies have demonstrated that high glucose levels significantly reduce the viability of RPE cells and induce pyroptosis in a manner that depends on both the duration and dosage of exposure [6,7]. In models of high-glucose-induced RPE, there is a notable increase in the expression of pyroptosis-related proteins, such as caspase-1 and NLRP3, along with elevated levels of inflammatory factors like IL-1β and IL-18 [79].

Non-coding RNAs participate in regulating high-glucose-induced RPE pyroptosis [80]. Huang et al. found that a circular RNA, circFAT1, was downregulated in RPE treated with high glucose, while the overexpression of circFAT1 led to enhanced expression of LC3B and reduced levels of GSDMD and inflammatory mediators, suggesting that circFAT1 inhibited RPE pyroptosis and promoted protective autophagy [81]. In addition, the knockdown of circZNF532 was shown to mitigate high-glucose-induced RPE pyroptosis via regulation of the miR-20b-5p/STAT3 signaling pathway [82]. Aberrantly high expression of lncRNA HOXD Cluster Antisense RNA 1 (HAGLR) was detected in RPE exposed to high glucose, and its knockdown alleviated RPE pyroptosis and cytotoxicity [83]. Overexpression of miR-192 in high-glucose-exposed RPE cells inhibited pyroptosis through the regulation of the FTO/NLRP3 signaling pathway [6]. Furthermore, targeting the miR-25-3p/PTEN/Akt signaling pathway through methyltransferase-like protein 3 [7] and the miR-20a/TXNIP axis through DNA methyltransferase 1 [79] showed potential therapeutic effects on high-glucose-induced RPE pyroptosis. Collectively, non-coding RNAs represent promising targets for regulating signaling pathways associated with high-glucose-induced RPE pyroptosis, although the further elucidation of their exact mechanisms is warranted.

Furthermore, Yumnamcha et al. found that auranofin, an inhibitor of thioredoxin reductase (TrxR), induced mitochondrial dysfunction and lactate dehydrogenase release in RPE, which could be reversed by NLRP3 inflammasome inhibitors, but they were not affected by inhibitors of ferroptosis or necroptosis. They suggested that the TrxR redox pathway may contribute to RPE dysfunction in DRN and other retinal neurodegenerative diseases [84].

#### 3.2.2. Retinal Ganglion Cells

RGC pyroptosis in DR has been reported, although research in this domain remains limited. Zhang et al. employed bioinformatics and network pharmacology approaches to identify key genes associated with RGC pyroptosis in DR. They found that salidroside significantly ameliorated RGC pyroptosis, potentially through the regulation of NLRP3, NFEZL2, and NGKB1 [85]. Another study indicated that the traditional Chinese medicine extract scutellarin (SCU) partially rescued RGCs from pyroptosis in DR by inhibiting caspase-1, GSDMD, NLRP3, IL-1β, and IL-18 [86]. These findings provide potential directions for targeting RGC pyroptosis in the treatment of DR. However, specific drugs designed to target RGC pyroptosis are currently lacking.

#### 3.2.3. Glia

The activation and proliferation of glial cells, along with the excessive release of inflammatory factors, are key events of neuronal inflammation [87]. Recent studies have revealed that glial cells undergo cell death within specific pathological microenvironments, which further exacerbates immune inflammation and tissue damage [88]. High glucose levels have been shown to impede retinal microglial function and enhance the expression of the pyroptotic core machinery, including caspase-1, GSDMD, NLRP3, and IL-1β, in a dose-dependent manner. Inhibition of the NLRP3 signaling pathway inhibited microglial cell pyroptosis and its mediated cytotoxicity [89]. Additionally, Müller cells undergo pyroptosis following retinal I/R injury, contributing to retinal neurodegeneration. This process was suppressed by blocking the NLRP3/GSDMD-N/Caspase-1/IL-1β pathway [90]. Ma et al. identified a novel EP300/H3K27ac/TRPC6 signaling pathway implicated in high-glucose-induced Müller cell pyroptosis, and knockdown of transient receptor potential channel 6 (TRPC6) significantly reduced inflammation and pyroptosis in Müller cells [91]. Glial cell pyroptosis could thus serve as a potential therapeutic target for DRN.

#### 3.2.4. Retinal Vascular Cells

High glucose induces the pyroptosis of RCECs. In the early stages of mouse models of DR, the P2X7/NLRP3 signaling pathway is activated, leading to pyroptosis of RCECs and a significant inflammatory response. Inhibition of the P2X7/NLRP3 axis reduced NLRP3 inflammasome-mediated damage [92,93]. Several non-coding RNAs have been identified as potential targets for pyroptosis of RCECs. The lncRNA HOTAIR was found upregulated in high-glucose-stimulated human RCECs, while the knockdown of HOTAIR inhibited the maturation, NLRP3 inflammasome activation, and pyroptosis [94]. In human RECEs treated with high glucose, miR-200c-3p was highly expressed [95], whereas miR-590-3p and miR-26a-5p were downregulated [96,97]. Targeting these microRNAs effectively inhibited the pyroptosis of RCECs [95,96,97].

Retinal microvascular pericyte loss is one of the earliest pathological changes associated with DR. Recent studies have suggested the involvement of pericyte pyroptosis in DR pathology. In vitro experiments have shown that high glucose induces NLRP3-caspase-1-GSDMD-dependent pericyte pyroptosis, leading to the release of inflammatory cytokines IL-1β, IL-18, and lactate dehydrogenase. This effect is dose- and time-dependent and can be blocked by caspase-1 or NLRP3 inhibition [98]. Similarly, in an environment mimicking DR created by advanced glycation end product-modified bovine serum albumin (AGE-BSA), retinal pericytes underwent caspase-1- and GSDMD-mediated active cleavage, along with the release of inflammatory IL-1β and IL-18. Treatment with the miR-342-3p mimic effectively inhibited pyroptosis in pericytes [99]. These findings provide new insights into early treatment strategies for DR.

In summary, cell pyroptosis represents a typical form of non-apoptotic PCD. Pyroptosis in RGCs directly leads to the impairment of retinal neural function. More importantly, the presence of pyroptosis in various cell types, including glial cells, exacerbates retinal neuroinflammation and retinal neurodegeneration under high-glucose conditions. Targeting specific components of pyroptosis signaling pathways, such as NLRP3 and caspase-1, may help modulate pyroptosis and alleviate the detrimental effects of inflammation and neuronal damage. Further evidence from in vivo experiments is needed to confirm these conclusions.

## 4. Necroptosis

### 4.1. Overview of Necroptosis

Necroptosis, first described by Alexei Degterev in 2005, is a form of PCD characterized by necrotic cell death morphology and activation of autophagy [100]. Morphologically, necroptosis is characterized by rapid plasma membrane penetration, the leakage of cell constituents, release of damage-associated molecular patterns, cell swelling, and mitochondrial membrane permeabilization [101]. Unlike passive necrosis, necroptosis is regulated by specific molecular signaling pathways. It is currently understood that necroptosis can be initiated by the ligation and activation of death receptors, including tumor necrosis factor receptor 1 (TNFR1) [102], Fas ligand (FasL), and TNF-related apoptosis-inducing ligand (TRAIL) [103], as well as pattern recognition receptors, such as toll-like receptor 3 (TLR3) [104] and TLR4 [105]. The activation of these receptors oligomerizes receptor-interacting protein kinase 1 (RIPK1) in the cytoplasm, leading to the formation of complexes with receptor-interacting protein kinase 3 (RIPK3), which is a critical step in the necroptosis signaling pathway. The RIPK1-RIPK3 complexes trigger the phosphorylation of the downstream mixed lineage kinase domain-like protein (MLKL) to form the “Necrosome”. MLKL then undergoes conformational changes and forms oligomers, leading to membrane permeabilization and cell lysis [102]. Subsequent research has identified additional signaling pathways that can induce necroptosis without RIPK1 activation, such as Z-DNA binding protein 1 (ZBP1) and TIR domain-containing adapter-inducing interferon-β (TRIF) [104,106]. Furthermore, RIPK3/MLKL-mediated necroptosis can also be inhibited by caspase-8 activation [107], indicating the versatility of necroptosis regulation.

Necroptosis is involved in a variety of disease conditions, including inflammation, infection, cancer, neurodegeneration, and others [108,109,110]. In ophthalmology, necroptosis is associated with the pathogenesis of diseases such as glaucoma, AMD, retinitis pigmentosa, retinal detachment, and DR. Neyra et al. showed the upregulation of pro-necroptotic genes and proteins in an in vitro model of retinal neurodegeneration [111]. They observed that MLKL expression began in the inner layers of the retina and progressed to the outer layers within 1 day, which was in accordance with photoreceptor degeneration [111]. Ma et al. demonstrated that excessive thyroid hormone signaling induced necroptosis in the retina, leading to photoreceptor degeneration [112]. Through gene expression analysis, Martin et al. showed that the secretome of human bone marrow mesenchymal stem cells could inhibit necroptosis activation in retinal neurodegeneration, suggesting necroptosis as a potential therapeutic target for retinal degenerative diseases [113].

### 4.2. Necroptosis in Diabetic Retinal Neurodegeneration

Similar to pyroptosis, necroptosis exacerbates neural damage through its inflammatory properties. In addition, the diabetic microenvironment fosters necroptosis in retinal neurons, leading to neurodegeneration and visual impairment. In vitro experiments have confirmed that exposure to hyperglycemic conditions triggers RIPK1- and MLKL-dependent necroptosis in various cell types. This process likely hinges on feedback mechanisms involving glycolysis, advanced glycation end products, and ROS, and can be blocked by RIPK1 inhibitors or siRNA interference [114]. The following section discusses the involvement of necroptosis in different cell types and its therapeutic promise in DRN.

#### 4.2.1. Retinal Ganglion Cells

Necroptosis has been demonstrated in cultured RGCs induced by high glucose, marked by the elevated expression of RIPK1/RIPK3. Treatment with the RIPK1 inhibitor Necrostatin-1 effectively protected RGCs from necroptotic cell death. Meanwhile, there was a significant increase in the number of Nissl bodies within the cells post-treatment, suggesting an improvement in cell function [115].

There are currently fewer reports confirming the presence of RGC necroptosis in DR at the animal level. Ischemia/hypoxia-induced ROS accumulation is one of the primary pathological changes in DR, and studies have found that in both oxygen-glucose deprivation (OGD) and I/R animal models, RGCs underwent RIPK3/MLKL-dependent necroptosis. Treatment with the RIPK3 inhibitor GSK840 alleviated neurodegeneration and improved retinal microstructure and visual function [116]. An in vitro study further confirmed the involvement of necroptosis under OGD, with RIPK3 assuming an important role in this process [117]. Elevated RIPK1/3 levels were also observed in the retina in an I/R mouse model [118], whereas GSK840 treatment improved electroretinogram amplitude and exerted protective effects on retinal neurons [116]. Activation of TNF-α was involved in the induction of RGC necroptosis in the I/R model, which could be blocked by Necrostatin-1 [119]. In addition, Gao et al. identified the extracellular signal-regulated kinase (ERK) 1/2 as a regulator of RIPK3-related necroptosis [120], while Lee et al. pinpointed Daxx as a downstream component of RIPK3 [121]. Targeting these molecules may aid in preventing necroptosis in the early stages of I/R injury.

#### 4.2.2. Microglia

Microglia, as pivotal mediators of neuroinflammation, are also subject to cell death during disease progression. Huang et al. highlighted the involvement of microglial necroptosis in exacerbating neuroinflammation and retinal neurodegeneration through RIPK1/RIPK3-dependent mechanisms, a process mitigated by Necrostatin-1 [105]. In a streptozotocin (STZ)-induced diabetes model, the use of the RIPK3 inhibitor GSK872 effectively inhibited microglial necroptosis and concomitant neuroinflammation, thereby rescuing the decrease in neuronal density and nerve-retinal thickness in diabetic mice [122]. In addition, He et al. identified a distinct microglia subpopulation, termed sMG2, in a hypoxia-induced retinal neovascularization model. The sMG2 subpopulation demonstrated specific activation of ripk3 and mlkl genes under hypoxic conditions, rendering them susceptible to necroptosis. The ensuring necroptosis induced the production of the angiogenic factor FGF2, contributing to retinal neovascularization [123]. These findings underscore the critical role of RIPK1/RIPK3-mediated necroptosis in microglia in diabetic neuroinflammation and neurodegeneration.

#### 4.2.3. Photoreceptors

Direct evidence linking necroptosis DRN in photoreceptors is still emerging. However, necroptosis does occur in photoreceptors under conditions mimicking DR pathology. RIPK3-dependent necroptosis was activated in H_2_O_2_-treated 661W cells, which could be blocked by the mitochondrially targeted peptide SS31, leading to protection against oxidative stress and cell death [124]. In a retinal detachment model, RIPK3 activation triggered necroptosis in photoreceptor cells, while the blockade of RIPK1 or deficiency of RIPK3 markedly abolished this effect [125]. Additionally, Sato et al. demonstrated the involvement of RIPK-mediated necroptosis in photoreceptor degeneration, indicating the potential role of the TNF-RIP pathway as a candidate target for retinal neurodegeneration [126].

In summary, necroptosis is a novel form of non-apoptotic PCD primarily driven by RIPK1 and/or RIPK3. During retinal neurodegeneration associated with conditions like diabetes, necroptosis predominantly affects RGCs and microglia and may also involve photoreceptors. Activation of necroptosis not only results in direct neuronal death but also triggers significant neuroinflammation, further fueling the neurodegenerative cascade. Several studies have demonstrated that blocking the necroptotic pathway through specific RIPK1/RIPK3 inhibitors or gene deletion can alleviate retinal neuroinflammation and neurodegeneration, indicating necroptosis as a promising therapeutic target for DRN.

## 5. Other Programmed Cell Death Types

### 5.1. Parthanatos

Parthanatos, defined by Dawson in 2008, is a form of non-apoptotic PCD dependent on poly (ADP-ribose) polymerase-1 (PARP1). This pathway is activated by oxidative stress-induced DNA damage, leading to excessive accumulation of poly (ADP-ribose) (PAR) and subsequent induction of cell death [127]. PAR accumulation induces the release of mitochondrial apoptosis-inducing factor (AIF), which translocates macrophage inhibitory factor (MIF) to the nucleus and cleaves genomic DNA into segments [127,128]. Parthanatos has been implicated in various neurodegenerative diseases, including Parkinson’s disease, diabetes mellitus, and cerebral I/R injury.

PARP1 plays a crucial role in parthanatos, which can lead to a significant decrease in NAD (+) and ATP depletion, ultimately resulting in cell death [127,128,129]. Knocking down PARP1 not only enhanced the viability of cultured human RCECs under high-glucose conditions but also prevented high-glucose-induced inflammation [130]. PARP inhibitors, which are currently undergoing clinical trials for cancer therapy, have shown potential in attenuating excitotoxicity and ischemic cell injury in neurons [131]. Diabetic peripheral neuropathy (DPN) is a common complication of diabetes characterized by neurovascular damage. Blocking high-glucose-induced oxidative stress suppressed parthanatos by reducing PAR accumulation and AIF nuclear translocation [132]. Overall, the use of PARP inhibitors holds promise as a neuroprotective therapy to reduce neuronal cell death and tissue damage.

Studies have shown that increased levels of excitotoxic glutamate in the retina may lead to retinal neuronal cell dysfunction or death through activation of the PARP1–parthanatos pathway. Erythropoietin has been identified as a potential neuroprotective agent in the diabetic retina by regulating glutamate levels and inhibiting parthanatos [133]. Additionally, nicotinamide, a form of vitamin B3, played a role in mitigating DRN by modulating the response to DNA damage. Treatment with niacinamide in diabetes led to a decrease in oxidative stress and cleaved PARP1 expression. This suggests that niacinamide may help mitigate DRN by promoting DNA repair [134]. While these findings from cellular and animal models are promising, there are still gaps in translating these neuroprotective treatments to clinical practice. Further research is needed to better understand the mechanisms underlying PARP1-mediated parthanatos in DRN and to develop effective clinical interventions.

### 5.2. PANoptosis

PANoptosis, a recently characterized mode of death involving the convergence of pyroptosis, apoptosis, and necroptosis, has emerged as a novel area of research in understanding cell death mechanisms. The innate immune sensor ZBP1 and TAK1 kinases play a role in regulating the assembly of the PANoptosis-like body complex [135]. Although the individual pathways of pyroptosis, apoptosis, and necroptosis are well studied, the interplay and regulation among them in PANoptosis remain complex and not fully understood. In the content of glaucomatous RGC damage, studies have investigated the involvement of PANoptosis. Treatment with melatonin has been shown to rescue RGC survival and reduce the loss of retinal nerve fiber layer thickness, possibly through inhibiting the expression of PANoptosis-associated proteins [136]. In addition, the inhibition of dynamin-related protein 1 (Drp1) demonstrated neuroprotective effects against high intraocular pressure-induced injury by regulating the expression of PANoptosis-associated proteins and the ERK1/2-Drp1-ROS pathway, suggesting a potential therapeutic strategy for RGC protection [137].

In retinal I/R injury models, the upregulation of PANoptosome components has been observed, accompanied by alterations in neuron morphology and protein levels, indicative of PANoptosis-like cell death [138]. Notably, studies investigating the protective effects of Dickkopf-1 in DR highlighted its role in inhibiting PANoptosis by blocking core proteins of pyroptosis, apoptosis, and necroptosis, as well as angiogenesis-related molecules, and endothelial cell proliferation and migration [139]. These results provide a basis for further investigation of this novel regulated cell death in DRN.

### 5.3. NETosis

NETosis is a form of PCD induced by neutrophil extracellular traps (NETs) [140]. NETs are released when the cell membrane is ruptured and are dependent on nicotinamide adenine dinucleotide phosphate (NADPH) oxidase for the production of ROS, leading to cell death and inflammatory response [141,142,143]. NETosis plays a critical role in the development of diabetes and its related complications. An elevated NET component has been observed in DR patients or mouse models of ocular inflammation, correlating positively with the severity of DR [143,144]. Notably, the activation of NADPH oxidase and production of ROS were implicated in high-glucose-induced NETosis, while anti-VEGF treatment was able to attenuate this process, suggesting a possible target in modulating NETosis in DR [143]. Binet et al. investigated the role of neutrophils in the remodeling of unhealthy vessels during advanced inflammation, showing that aging vasculature attracted neutrophils and induced the production of NETs [145]. These findings suggest that NETosis is primarily involved in vasculopathy rather than neurodegeneration in DR.

Furthermore, NETosis has been implicated in DPN with histone deacetylase playing a crucial role in its progression. Suppressing histone deacetylase has been shown to inhibit NETosis, reduce DRN, and alleviate pain associated with DPN [146]. The association of NETosis with neurodegeneration in the retina remains to be further investigated.

### 5.4. Other Non-Apoptotic PCD

Autosis, a form of PCD triggered by excessive or uncontrolled levels of autophagy [147,148], has been studied in the context of neuronal death after cerebral hypoxia-ischemia injury [148]. While its role in some diabetes-related complications has been explored, its involvement in DR remains poorly understood. Nonetheless, considering its potential role in hypoxia-ischemia-induced neurotoxicity, autosis may represent a promising neuroprotective target in DRN.

Lysosomal cell death (LCD) is a non-apoptotic PCD characterized by lysosomal rupture, lipid metabolites, and ROS production mediated by cathepsins or iron [15,19]. LCD has implications in inflammation, neurodegeneration, and aging [18]. Lysosomal dysfunction can impact neurons, and oxidative stress may contribute to neurodegenerative diseases [149]. Das et al. suggested the intervention of the endo-lysosomal pathway as a therapeutic approach for optic neuropathies [150]. However, the molecular mechanisms underlying this form of PCD in the context of DRN are currently unknown and require further investigation.

Other recently identified non-apoptotic forms of PCD include entotic cell death (Entosis), alkaliptosis, and oxeiptosis. Entosis involves cellular “cannibalism” where one cell engulfs and kills another [151]. Alkaliptosis is driven by intracellular alkalization, induced by the downregulation of the NF-κB-dependent carbonic anhydrase 9 (CA9) [152]. Oxeiptosis is a caspase-independent PCD induced by ROS, potentially relevant to DR pathology [153], although its specific involvement in DRN is minimal.

Moreover, cuproptosis is a newly discovered type of PCD associated with an imbalance in intracellular copper metabolism. Excess copper can lead to neurodegenerative diseases by causing protein-toxic stress reactions [16]. Disulfidptosis is another novel PCD involved in sulfur metabolism. SLC7A11, a regulator of redox homeostasis, plays a role in the induction of disulfide stress and rapid cell death [17]. However, limited studies have explored these modes of PCD in DR and their implications for neuroprotection. Further exploration is warranted, which may open a new research direction for DR and its neuroprotective mechanisms.

## 6. Conclusions and Perspectives

We summarized several non-apoptotic PCD mechanisms and potential pharmacological targets in the context of DRN, including ferroptosis, pyroptosis, necroptosis, and the emerging PANoptosis (Appendix A). Neurodegeneration is a critical pathology in the early phases of DR, driven by inflammation, oxidative stress, neuronal death, and I/R injury. Multiple PCD types coexist in DRN, each playing a distinct pathogenic role: pyroptosis predominantly mediates inflammation, while ferroptosis induces oxidative stress damage. Future investigations should focus on identifying key determinants of retinal cell death types under diabetic conditions and their interrelationships within DRN. This is crucial for the development of PCD-targeted therapies. Furthermore, a comprehensive understanding of PANoptosis’s mechanistic role in DRN may aid in creating integrative therapeutic strategies. Pathways like NETosis, autosis, and LCD are insufficiently explored in diabetic neurodegenerative diseases. By uncovering these molecular mechanisms and key signaling pathways, we aim to develop more precise and effective treatment strategies for DRN patients, thereby improving their quality of life.

## Figures and Tables

**Figure 1 pharmaceuticals-17-00837-f001:**
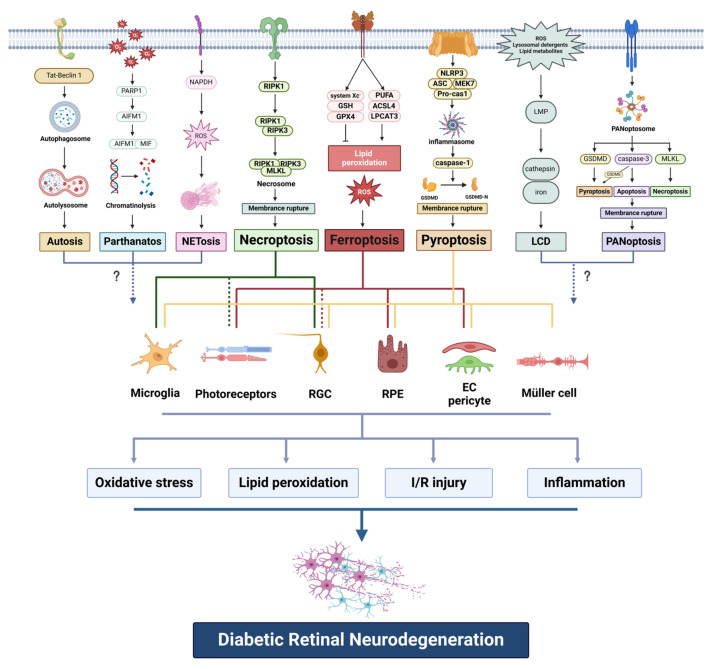
Molecular mechanisms of non-apoptotic programmed cell death in diabetic retinal neurodegeneration.PARP1: poly (ADP-ribose) polymerase-1; AIFM1: apoptosis inducing factor mitochondria associated 1; MIF: macrophage migration inhibitory factor; NAPDH: nicotinamide adenine dinucleotide phosphate; ROS: reactive oxygen species; RIPK1: receptor-interacting protein kinase 1; RIPK3: receptor-interacting protein kinase 3; MLKL: mixed lineage kinase-like protein; GSH: glutathione; GPX4: glutathione peroxidase 4; PUFA: polyunsaturated fatty acid; ACSL4: acyl-CoA synthetase long chain family member 4; LPCAT3: lysophosphatidylcholine acyltransferase 3; NLRP3: nod-like receptor family pyrin domain containing 3; ASC: adaptor protein-apoptosis-associated speck-like protein containing a CARD; MEK7: mitogen-activated protein kinase extracellular signal-regulated kinase 7; Pro-cas-1: pro-caspase-1; GSDMD: gasdermin D; LMP: lysosomal membrane permeability; LCD: lysosome-dependent cell death; GSDME: gasdermin E; RGC: retinal ganglion cell; RPE: retinal pigment epithelium; EC: endothelial cell; I/R: ischemia-reperfusion. Dotted lines and question marks indicate that the correlation is not yet fully established.

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
