# Peer review of "Non-Apoptotic Programmed Cell Death as Targets for Diabetic Retinal Neurodegeneration"

_pharmaceuticals, 2024, doi:10.3390/ph17070837_

Round 1

Reviewer 1 Report

Comments and Suggestions for Authors

The paper entitled “Non-apoptotic Programmed Cell Death as Targets for Diabetic 

Retinal Neurodegeneration” by Lin et al. is a review on DR damage and cell death.

This review aims to highlight recent advancements in identifying the role of distinct non-apoptotic PCD types across diverse retinal cell populations within the diabetic retina.

This paper is interesting, however the authors may wish to consider the following prior to publication.

The retina, just like the brain, becomes old and increases its susceptibility to age-related neurodegenerative diseases—glaucoma, diabetic retinopathy and AMD. Age-related retinal dysfunction results from structural and functional changes among which there are vascular defects linked to small vessel damage, elevated blood pressure, other microvascular processes, as well as failures in microglial motility, branching and length. Interestingly, changes in neurosteroids can impact the protection of retinal tissues.

Estradiol, allopregnanolone, dehydroepiandrosterone-sulfate has been suggested to provide positive effects in diseases’ models where neurosteroid synthesis is affected. Further, it has been suggested that neurosteroids may affect the metabolic state of surviving neurons and glial cells after ischemic injury and that they act through involvement of sigma1 recognition sites (please report the relevant paper: European Journal of Pharmacology 498 (2004) 111–114). It has been demonstrated that sigma-1receptor regulates the cell death (including pyroptosis) induced by brain injury (J Clin Med 2022, 11, 2348); further  sigma-1receptor ligands protect human retinal cells against oxidative stress that occurs during diabetic retinopathy (please report this study: NeuroReport 2006, 17, 287-291). The authors should add the above comment in the paragraph 3.1.

Comments on the Quality of English Language

Moderate editing of English language required

Author Response

Response:

We sincerely appreciate the valuable comments provided by the reviewer. As the reviewer mentioned, neurosteroids may affect the metabolic state of surviving neurons and glial cells after ischemic injury and act through involvement of sigma1 recognition sites. According to the literatures, the Sigma-1 receptor is a transmembrane protein located on the endoplasmic reticulum and mitochondria-associated membranes (MAM). It plays a crucial role in regulating cell survival, death, and stress responses. The Sigma-1 receptor prevents pyroptosis and relevant inflammation by regulating endoplasmic reticulum stress and calcium signaling. It also protects retinal cells from oxidative stress damage and regulates cell survival mechanisms, playing a protective role in the pathology of diabetic retinopathy. Therefore, it may be a potential target for intervention in diabetic neurodegeneration. We have included all the above content in the revised manuscript (paragraph 3.1; Page 7, Line 295-304).

Besides, we have critically revised the manuscript and added substantial information per the reviewers’ suggestions. In particular, the entire manuscript has been reviewed and polished by Dr. M Brelen, one of our native English-speaking colleagues.

Reviewer 2 Report

Comments and Suggestions for Authors

This is a very interesting review concerning the evaluation of non-apoptotic programmed cell death as targets for diabetic retinal neurodegeneration.

This is a well-written and well-organized review, discussing a very challenging topic. The Figure 1 is very pertinent, as well as the organization of the manuscript.

Also the related references are good.

I only suggest the authors to add a Table (or more than one) where all the non-apoptotic programmed cell death types across diverse retinal cell populations are summarized.

Author Response

Response:

We sincerely appreciate the reviewer for acknowledging our paper an interesting and well-organized one. Per the reviewer’s suggestion, we now add a Table to summarize the non-apoptotic programmed cell death types across diverse retinal cell populations in diabetic retinal degeneration (Table 1). In addition, the entire manuscript has been reviewed and polished by Dr. M Brelen, one of our native English-speaking colleagues.

Reviewer 3 Report

Comments and Suggestions for Authors

Amygdalin is an isolated substance, therefore it cannot be a “traditional medicine” at best an intredient of traditional medicines.

The most important deficit of this paper is the fact that is devoid of any visual support. Several figures could be useful in providing a more synthetic and clear image of the main forms of CPD and the signalling pathways involved.

“Das suggested that intervention of  endo-lysosomal pathway may provide therapeutic protection…”. What is “Das” in this context?

Comments on the Quality of English Language

Only minor editing is necessary.

Author Response

Response:

We agree with the reviewers' comments that figures can represent scientific content in a more vivid, intuitive, and concise manner. In this paper, we have attempted to use a figure to provide an overall summary of the different forms of PCD and their classic signaling pathways. We have now added a new table to summarize the potential roles of PCD in diabetic retinopathy, making it easier for readers to grasp the main information of the paper.

We apologize for the confusion caused by not expressing this clearly. “Das” in the phrase “Das suggested that intervention of endo-lysosomal pathway may provide therapeutic protection…” refers to the author's name, Arupratan Das. We now revise it. (Page 13, Line 589)

Besides, the entire manuscript has been reviewed and polished by Dr. M Brelen, one of our native English-speaking colleagues.

Round 2

Reviewer 1 Report

Comments and Suggestions for Authors

Dear authors,

the new version of the manuscript can be accepted now.

Comments on the Quality of English Language

Minor editing of English language required